# Feasibility and Safety of Uniportal Thoracoscopic Segmentectomy Using a Unidirectional Dissection Approach without Dissecting a Fissure

**DOI:** 10.3390/medicina60060994

**Published:** 2024-06-17

**Authors:** Hitoshi Igai, Mitsuhiro Kamiyoshihara, Kazuki Numajiri, Fumi Ohsawa, Kazuhito Nii

**Affiliations:** Department of General Thoracic Surgery, Japanese Red Cross Maebashi Hospital, 389-1 Asakura-cho, Maebashi 371-0811, Gunma, Japan; micha2005jp@yahoo.co.jp (M.K.); kznumajiri@gmail.com (K.N.); f.shiraishi0827@gmail.com (F.O.); knii2121@gmail.com (K.N.)

**Keywords:** uniportal, thoracoscopy, segmentectomy, unidirectional dissection

## Abstract

*Background*: Few original articles describe the perioperative outcomes of uniportal thoracoscopic segmentectomy using a unidirectional dissection approach. In this retrospective study, we evaluated the feasibility and safety of this procedure. *Methods*: This study included 119 patients who underwent uniportal thoracoscopic segmentectomy in our department between February 2019 and December 2022. The patients were divided into unidirectional (group U, n = 28) and conventional (group C, n = 91) dissection approach groups. While the dominant pulmonary vessels and bronchi were transected at the hilum without dissecting a fissure in the unidirectional (U) group, the dominant pulmonary artery was exposed and divided at a fissure in the conventional (C) group. Patient characteristics and perioperative outcomes were compared between groups U and C. *Results*: The proportions of simple and complex segmentectomies were statistically similar between the groups. The operating time was shorter (group U: 110 [interqurtile range: 90–140] min, group C: 135 [interqurtile range: 105–166] min, *p* = 0.012) and there was less blood loss (group U: 0 [interqurtile range: 0–0] g, group C: 0 [interqurtile range: 0–50] g, *p* = 0.003) in group U than in group C. However, there were no significant intergroup differences in other perioperative outcomes. *Conclusions*: The unidirectional dissection approach in uniportal thoracoscopic pulmonary segmentectomy is safe and feasible and enables a smoother operation.

## 1. Introduction

The number of patients undergoing pulmonary segmentectomy is expected to increase in patients with stage IA non-small-cell lung cancer (NSCLC) [1,2]. Therefore, we believe that it is necessary to perform uniportal thoracoscopic segmentectomy even via uniport if it is adopted as a standard approach in the institution, although it is considered technically more difficult than a multiport approach because the angulation of the inserted surgical instruments is limited by the single small skin incision [3,4,5,6].

To overcome the technical difficulties, our team adopted a “unidirectional dissection approach” for segmentectomy employing a uniportal thoracoscopic method [7,8]. With unidirectional dissection, the dominant pulmonary veins and arteries, and bronchi, are divided sequentially at the hilum, followed by division of the fissure. This approach is similar to the “fissureless technique” in lobectomy [9,10,11,12]. Therefore, it is not affected by the fissure grade, whereas most types of segmentectomy require dissection of a fissure to expose the pulmonary arteries, which is considered a conventional technique. However, a disadvantage of this approach is that the unidirectional surgical view is sometimes complicated and requires a thorough understanding of the anatomy. Therefore, it may occasionally be difficult to achieve a proper segmentectomy using the unidirectional dissection approach, particularly for less experienced surgeons.

Few original articles describe the perioperative outcomes of uniportal thoracoscopic segmentectomies using a unidirectional dissection approach because it has not been well known [13,14]. In addition, only a few authors have presented the surgical steps in case reports [7,8,15]. In this retrospective study, we evaluated the feasibility and safety of uniportal thoracoscopic segmentectomy using a unidirectional dissection approach by comparing the perioperative outcomes with using a conventional approach.

## 2. Patients and Methods

The study was approved by the institutional ethics committee of the Japanese Red Cross Maebashi Hospital (approval number: 2023–1, date: 26 April 2023). The need for individual consent was waived for this retrospective analysis.

We enrolled 119 patients who underwent uniportal thoracoscopic segmentectomy in our department between February 2019 and December 2022. The patients were divided into unidirectional (group U, n = 28) and conventional (group C, n = 91) dissection approach groups. While the dominant pulmonary vessels and bronchi were transected at the hilum without dissection of a fissure in group U, the dominant pulmonary artery was exposed and divided at a fissure in group C. Groups U and C were compared in terms of patient characteristics and perioperative outcomes. Figure 1 shows the patient enrollment process.

The clinical data analyzed for each patient included age, sex (male or female), lobe containing resected segment (left upper, left lower, right upper, right middle, or right lower), American Society of Anesthesiologists (ASA) score, smoking index (pack-years), forced expiratory volume in 1 s (FEV1.0), %FEV1. 0, disease (primary lung cancer, pulmonary metastasis, or other benign), kind of segmentectomy (intentional, unintentional, or other), operating time, blood loss, postoperative drainage time, postoperative hospital stay, morbidity (Clavien–Dindo grade ≥ III), readmission within 30 days after surgery, conversion to thoracostomy, and 30-day postoperative mortality.

At our institution, uniportal thoracoscopic pulmonary segmentectomy is planned for confirmation or removal of a malignancy. For cTis-1aN0M0 primary lung cancer, intentional segmentectomy with lymph node sampling was performed with patient consent. Patients with poor pulmonary function or cardiopulmonary status underwent an unintentional segmentectomy as a passive limited resection. Wedge resection was generally chosen for patients with pulmonary metastases from other cancers. Segmentectomy was performed when the tumor location made it difficult to achieve a safe margin with wedge resection. The same criteria as for pulmonary metastasis were used to select the surgical procedure for benign disease. All segmentectomies were classified into simple and complex types [16,17]. Simple pulmonary segmentectomy included the lingual, basilar, or superior segment of the lower lobe or the upper division of the left upper lobe. Complex pulmonary segmentectomy was defined as any pulmonary segmentectomy other than those mentioned above. The surgery was performed by two senior and three junior surgeons. Each surgery was supervised by HI, which ensured the quality of the surgery.

### 2.1. Preoperative Assessment

Multidetector row computed tomography (CT) was used to localize the tumor. To confirm the branching pattern of the pulmonary vessels and bronchi, three-dimensional CT broncho-angiography (3DCTBA) was performed in all patients, except those with contrast allergy. Figure 2 shows the correspondence between the preoperatively simulated and actual branching patterns of the pulmonary vessels and bronchi. Ziostation2 software was used to simulate the virtual intersegmental plane preoperatively to ensure a sufficient surgical margin [18,19].

### 2.2. Surgical Procedure

Surgery was performed under general anesthesia with single-lung ventilation and the patient in the lateral decubitus position. Using a uniportal thoracoscopy approach, a single 3.5–4 cm skin incision was made on the anterior axillary line of the fourth or fifth intercostal space and initially covered with an extra-small Alexis retractor (Applied Medical, Rancho Santa Margarita, CA, USA). The incision was used to insert all surgical instruments and a 10 mm, 30-degree angled thoracoscope. The single skin incision and operative findings are shown in Figure 3. Large vessels or bronchi were divided using stapling. Small-caliber vessels were divided with an energy device after ligation. After completing the pulmonary segmentectomy, the specimen was placed in an endovascular bag and retrieved through the access incision. No rib spreader was used in any case. In patients with primary lung cancer undergoing segmentectomy, interlobar and hilar lymph nodes were sampled to confirm the pathological stage. If a resected lymph node was positive in an intentional segmentectomy, we planned to perform an additional lobectomy. For unintentional segmentectomies, no additional lobectomy was planned, as these patients could not tolerate an additional lobectomy. Patients with metastatic or benign disease did not undergo lymphadenectomy.

To identify the intersegmental plane accurately, we used the inflation–deflation technique or infrared thoracoscopic observation with intravenous indocyanine green (ICG) [20], using the intersegmental pulmonary veins in the hilum area as landmarks. The intersegmental plane was divided with staplers after identification. Finally, the intersegmental plane was covered with a sheet of polyglycolic acid (Neovail, sheet type; Gunze, Kyoto, Japan), and fibrin glue spray (Beriplast P; CSL Behring, King of Prussia, PA, USA) was applied to the intersegmental plane when the intraoperative sealing test showed air leakage in the divided intersegmental plane.

### 2.3. Example of the Surgical Procedure Using a Unidirectional Dissection Approach

As examples, we have provided two videos of surgical procedures using the unidirectional dissection approach.

Upper division segmentectomy of the left upper lobe.

The details are shown in Appendix A. First, we dissected the hilum. Then, V3, V1 + 2, and the pulmonary artery branches (A3, A1 + 2a + b, and A1 + 2c) were divided in sequence. After division, the upper divisional segmental bronchus (B1–3) was exposed, encircled, and divided with a stapler. When inserting the stapler, care was taken not to injure the pulmonary arteries behind the upper divisional bronchus. During these procedures, the upper lobe was always retracted in a caudal direction with forceps.

Infrared thoracoscopic observation with intravenous ICG was used to identify the intersegmental plane between the upper division and lingual segments. The identified demarcation line was marked by electrocautery. Finally, the intersegmental plane between the superior division and lingual segments was divided from anterior to posterior with staplers, maintaining a sufficient surgical margin.

Left lateral and posterior basal segmentectomy.

Details are shown in Appendix A. Throughout the procedure, the lower lobe was always retracted toward the head using forceps. We first incised the pulmonary ligament up to the inferior pulmonary vein and the posterior mediastinal pleura. Then, V9 + 10 was divided with a stapler. Next, the lateral and posterior basal segmental bronchus (B9 + 10) was completely exposed and divided with a stapler. After this, A9 + 10 was exposed and divided with a stapler, retracting the distal bronchial stump toward the head, which allowed us to dissect A9 + 10 easily. To create space for the stapler, which divided the intersegmental plane, the hilar tissue was dissected. The plane between S9 + 10 and the other segments was identified via infrared thoracoscopic observation after intravenous administration of ICG and divided with the stapler.

### 2.4. Postoperative Course

The chest tube was removed after confirming that there was no active bleeding and no air leak. From February 2019 to June 2021, the tube was left in at least until postoperative day 1. However, beginning in July 2021, we started removing the chest tube on the day of surgery [21]. Patients were discharged if a chest X-ray taken the day after chest tube removal showed no problems. Postoperative complications were evaluated according to the Clavien–Dindo classification [22].

### 2.5. Statistical Analysis

The Mann–Whitney *U*-test for continuous variables or Fisher’s test exact for categorial variables was used to compare patient characteristics and perioperative results between the two approaches. Differences were considered significant at *p* < 0.05. All calculations and statistical analyses were performed using the EZR graphical user interface for R (Saitama Medical Centre, Jichi Medical University, Saitama, Japan).

## 3. Results

### 3.1. Patient Characteristics

The patient characteristics between groups U and C are compared in Table 1. Most characteristics did not significantly differ between the two groups. However, the distribution of the lobe containing resected segment did (*p* < 0.001): upper lobes accounted for 82% of group U but only 48% of group C.

### 3.2. Distributions of the Performed Segmentectomies in Both Groups

The details of segmentectomies performed using the two different approaches are compared in Table 2. There were 12 simple (43%) and 16 (57%) complex types in group U versus 41 simple (45%) and 50 (55%) complex types in group C, which did not significantly differ. In left S3 segmentectomy and right S1 segmentectomy, we always used a unidirectional dissection approach. Therefore, group C did not include the patients who underwent these types of segmentectomies. On the contrary, all patients who underwent left S1 + 2 segmentectomy, right S2 segmentectomy, or S6 segmentectomy were included in group C because our uniportal approach, which placed the single skin incision at the anterior axillary line, cannot provide a good surgical view in these types of segmentectomies.

### 3.3. Perioperative Outcomes

The perioperative outcomes between the groups are compared in Table 3. The operating time was shorter in group U than in group C (group U: 110 min, IQR: 90–140 min, group C: 135 min, IQR: 105–166 min, *p* = 0.0124), as was the intraoperative blood loss (group U: 0 g, IQR: 0–0 g, group C: 0 g, IQR: 0–50 g, *p* = 0.0034). There were no significant differences in other perioperative outcomes including postoperative drainage and hospitalization times.

### 3.4. Subset Analyses Excluding Left S1 + 2, S3, S6 and Right S1, S2, S6 Segmentectomies

In addition, we performed subset analyses excluding left S1 + 2, S3, S6 and right S1, S2, S6 segmentectomies. Appendix A outline the patient characteristics, details of the segmentectomies performed, and the perioperative outcomes in the subset analysis. In patient characteristics, although the distribution of the lobe containing resected segment was statistically different, other variables were similar. The proportion of simple-type segmentectomy in group U was significantly higher than that in group C (group U: 70.6%, group C: 40.8%, *p* = 0.049). The operating time was shorter in group U than in group C (group U: 123 min., IQR: 100–130 min., group C: 151 min., IQR: 120–180 min, *p* = 0.0033), as was the intraoperative blood loss (group U: 0 g., IQR: 0–0 g., group C: 0 g., IQR: 0–50 g, *p* = 0.0024).

## 4. Discussion

This study demonstrated that the unidirectional dissection approach can be performed safely in uniportal thoracoscopic segmentectomy with thorough knowledge of the anatomy. Most of the perioperative outcomes in the unidirectional dissection approach group were acceptable and not significantly different from the conventional approach. In addition, the operating time was significantly shorter with the unidirectional dissection approach, indicating that surgery was smoother with this approach. We attribute this to avoiding dissection of a dense fissure and avoiding repeated lung turnover. Previously, Liu et al. [14]. reported the same advantages of this approach. However, their research was a single-arm study that only included single or combined basal segmentectomy. We showed that this unidirectional dissection approach is useful for both lower and upper lobe segmentectomy. Subset analysis excluding left S1 + 2, S3, S6 and right S1, S2, S6 segmentectomies that were only included in one of the two groups also showed significantly shorter operative times in group U.

With our unidirectional dissection approach, the target pulmonary arterial branches were not treated at the fissure but at the hilum. This sometimes leads to misunderstanding of the anatomy, as the branching pattern of the pulmonary arteries and bronchus must be identified in the opposite direction to the conventional interlobar approach. To avoid this, a careful preoperative 3DCTBA simulation is important for understanding the relative positions of the pulmonary vein, bronchus, and pulmonary artery, although it is also useful for the conventional approach. The recent ESTS expert consensus report also recommended the use of 3D reconstruction as a significant aid in identifying the branching pattern of the segmental hilar vessels and bronchi [23]. In this report, they also recommended that the pulmonary vein be divided last and at the most peripheral site possible to prevent the division of a segmental vein that also drains another segment. Furthermore, in thoracoscopic pulmonary segmentectomy, which we described previously [18], this preoperative simulation using infrared thoracoscopic observation with ICG administration helped to ensure a sufficient surgical margin. Therefore, in our department, every patient who underwent thoracoscopic segmentectomy had 3DCTBA preoperatively, except for those with contrast allergy.

The unidirectional dissection approach has the advantage of not being affected by the fissure grade, which avoids postoperative prolonged air leakage (PAL), even in patients with a dense fissure. For such patients, several authors have reported the efficacy of the fissureless technique in lobectomy, which is similar to the unidirectional dissection approach in segmentectomy, to reduce the postoperative drainage time and the incidence of PAL. Ng et al. reported that the fissureless technique significantly reduced the rate of PAL, postoperative drainage time, and postoperative hospital stay in right upper lobectomy compared to the conventional approach [10]. In a randomized controlled trial that included all types of lobectomies, Gómez-Caro et al. also reported favorable results for the fissureless technique [11]. Stamenovic et al. showed that the thoracoscopic fissureless lobectomy group had significantly less PAL than conventional thoracoscopic due to the chest tube drainage duration and hospital stay with an equivalent operating time [12]. Moreover, our group insists on the usefulness of fissureless lobectomy for a dense fissure, because a dense fissure is usually a major cause of postoperative PAL [9]. The present study also showed excellent results for the postoperative drainage time in the patients who underwent unidirectional dissection, although it was not significantly different from those who underwent conventional dissection. Our results indicate that not dissecting such a dense fissure is crucial for avoiding PAL, not only in lobectomy but also in segmentectomy.

The distribution of segmentectomies performed differed between the two groups. In unidirectional dissection, the direction of dissection was always from anterior to posterior because our single skin incision was made on the anterior axillary line. However, it might be technically difficult to obtain good surgical view from the posterior to the anterior side even if 30-degree-angled thoracoscopy was rotated. In particular, it was expected that we would have difficulty exposing the pulmonary artery because it was located in the central part of the hilum covered by the lung. Therefore, posterior segments such as S2, S1 + 2, and S6 are not good candidates for this approach, and group U did not include these segmentectomies. Fortunately, in each case that had this type of segmentectomy, a fissure was clearly separated, which allowed us to expose the dominant pulmonary arteries at the fissure. However, if we encounter a dense fissure and want to use the unidirectional approach, a good option may be to change the position of the single skin incision followed by a posterior-to-anterior dissection.

In pulmonary segmentectomy, it is most important to achieve a sufficient surgical margin and ultimately avoid local recurrence. Compared to lobectomy, segmentectomy tends to lead to local recurrence because the surgical margin is close to the target tumor when the tumor is located near the intersegmental plane. In a previous study, among patients with NSCLC located 2 cm or less peripherally, a segmentectomy group had a local recurrence rate of 10.5%, while a lobectomy group had only 5.4%, which significantly differed (*p* = 0.0018), although the overall survival significantly favored the segmentectomy group [1]. We did not report the local recurrence rate because it included NSCLC and diseases; we only report short-term perioperative results. A future study should evaluate whether the type of surgical procedure, including unidirectional and conventional dissection, affects the local recurrence rate.

### Limitations

The current study had several limitations. First, it involved a single institution and was retrospective in nature with a small number of enrolled patients. Second, multiple surgeons were involved, which could affect the perioperative outcomes, although each operation was supervised by HI. Third, the distribution of the performed segmentectomies differed between the two groups, which might have caused the significant difference in the operative time between the two groups. In addition, the emergence of new equipment and the presence of different surgical nurses or assistants may affect perioperative outcomes. Finally, long-term outcomes, including the local recurrence rate, were not evaluated.

## 5. Conclusions

Unidirectional and conventional dissection approaches in uniportal thoracoscopic pulmonary segmentectomy had equivalent perioperative outcomes, and the unidirectional dissection approach had a shorter operating time despite the bias in resected segments. Therefore, the unidirectional dissection approach in uniportal thoracoscopic pulmonary segmentectomy is safe and feasible and allows smoother operation than the conventional approach by avoiding dissection, even in a dense fissure, and unnecessary repeated turning of the lung. Although posterior segments such as S2, S1+2, and S6 are not good candidates for this approach, it can be performed in any other segmentectomies regardless of the fissure grade. Careful preoperative simulation using 3DCTBA is considered a key factor to performing this procedure appropriately.

## Figures and Tables

**Figure 1 medicina-60-00994-f001:**
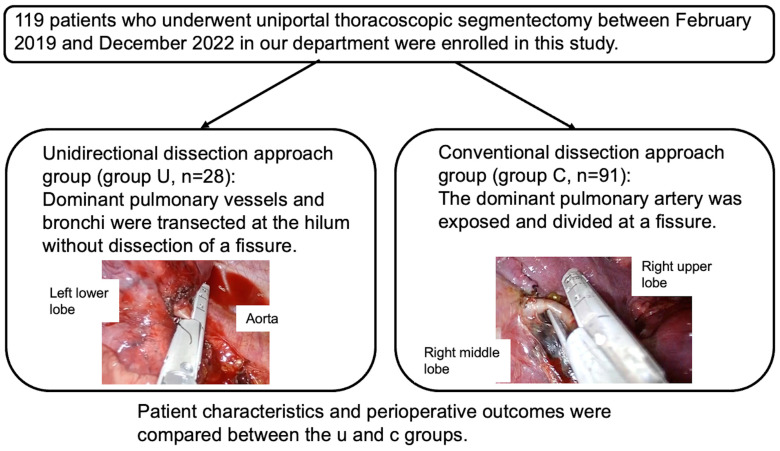
Patient enrollment.

**Figure 2 medicina-60-00994-f002:**
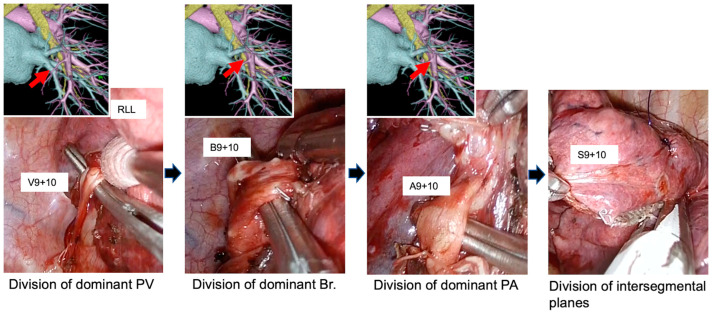
Correspondence between preoperatively simulated and actual branching patterns of pulmonary vessels and bronchi during uniportal thoracoscopic right lateral and posterior basal segmentectomy using a unidirectional dissection approach. During the procedures, the lower lobe was always retracted toward the head with forceps. The dominant pulmonary veins and arteries, and bronchi, were divided sequentially at the hilum without dissecting a fissure. The relative intraoperative positions of these structures corresponded to the simulated preoperative positions. After division of these structures, intersegmental planes were divided with staplers.

**Figure 3 medicina-60-00994-f003:**
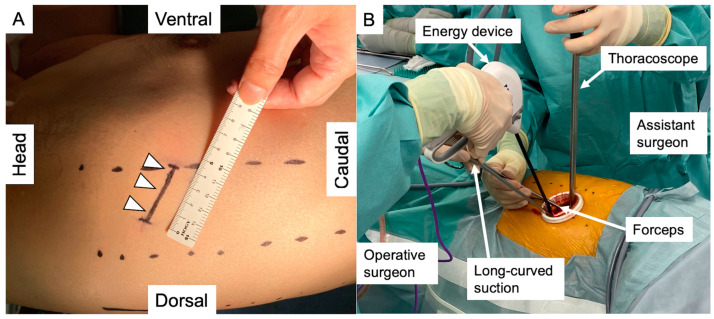
(**A**) A 3.5–4 cm single skin incision (arrowheads) was made at the 4th or 5th intercostal space of the anterior axillary line. (**B**) The operating surgeon stood at the ventral side of the patient, while an assistant stood at the dorsal side. The operating surgeon mainly used an energy device to dissect the intrathoracic structures and retracted the dissected area with a long, curved suction apparatus. An assistant surgeon manipulated a 30-degree thoracoscope and retracted the lung using forceps.

**Table 1 medicina-60-00994-t001:** Comparison of patient characteristics between groups U and C.

	Group U (n = 28)	Group C (n = 91)	*p*-Value
Age, years, median (IQR)	74 (68–79)	70 (66–77)	0.143
Sex Female/Male, n (%)	15 (54)/13 (46)	41 (45)/50 (55)	0.52
Lobe containing the resected segment LUL, n (%) LLL, n (%) RUL, n (%) RML, n (%) RLL, n (%)	14 (50) 4 (14) 9 (32) 0 (0) 1 (4)	20 (22) 14 (15) 24 (26) 0 (0) 33 (36)	<0.001
ASA score, median (IQR)	2 (2–2)	2 (2–2)	0.71
Smoking index, pack × year, median (IQR)	30 (3–49)	11 (0–40)	0.043
Preoperative FEV1.0, mL, median (IQR)	1870 (1578–2183)	1995 (1698–2628)	0.21
Preoperative %FEV1.0, %, median (IQR)	83 (73–97)	93 (78–107)	0.091
Disease Primary lung cancer, n (%) Pulmonary metastasis, n (%) Other benign, n (%)	19 (68) 4 (14) 5 (18)	65 (71) 10 (11) 16 (18)	0.89
Kind of segmentectomy Intentional, n (%) Unintentional, n (%) Others, n (%)	13 (46) 8 (29) 7 (25)	47 (52) 18 (20) 26 (29)	0.63

IQR, interquartile range; LUL, left upper lobe; LLL, left lower lobe; RUL, right upper lobe; RML, right middle lobe; RLL, right lower lobe; ASA, American Society of Anesthesiologists; FEV, forced expiratory volume.

**Table 2 medicina-60-00994-t002:** Details of segmentectomies performed in groups U and C.

	Group U (n = 28)	Group C (n = 91)	*p*-Value
Simple/Complex, n (%)	12 (43)/16 (57)	41 (45)/50 (55)	1
LUL S1–3 S1 + 2 S3 S3–5 S4–5	11 0 4 0 1	7 6 0 2 4	
LLL S6 S8 S8–10 S9–10 S10	0 0 0 3 0	8 4 1 0 1	
RUL S1 S1 + 3 S2 S2 + 1a S3 S2 + 6	7 0 0 0 1 0	0 3 11 2 7 1	
RLL S6 S7–8 S7–9 S7–10 S8 S8–10 S9–10	0 0 0 0 0 0 1	14 1 1 7 1 2 7	

LUL, left upper lobe; LLL, left lower lobe; RUL, right upper lobe; RLL, right lower lobe.

**Table 3 medicina-60-00994-t003:** Comparison of perioperative outcomes between groups U and C.

	Group U (n = 28)	Group C (n = 91)	*p*-Value
Operating time, min, median (IQR)	110 (90–140)	135 (105–166)	0.012
Blood loss, g, median (IQR)	0 (0–0)	0 (0–50)	0.003
Postoperative drainage time, days, median (IQR)	1 (0–1)	1 (1–1)	0.41
Postoperative drainage time 0–1, days (%) 2+, days (%)	23 (82) 5 (18)	78 (86) 13 (14)	0.76
Postoperative hospitalization time, days, median (IQR)	2 (2–3.5)	2 (2–3)	0.43
Morbidity (Clavien–Dindo classification grade ≥ 3), n (%)	0 (0)	5 (6)	0.59
Readmission within 30 days after discharge, n (%)	0 (0)	1 (1)	1.00
Conversion to thoracotomy, n (%)	0 (0)	4 (4)	0.57
30-day mortality, n (%)	0 (0)	1 (1)	1.00

IQR, interquartile range.

## Data Availability

The data underlying this article will be shared upon reasonable request to the corresponding author.

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
