# Peer review of "Feasibility and Safety of Uniportal Thoracoscopic Segmentectomy Using a Unidirectional Dissection Approach without Dissecting a Fissure"

_medicina, 2024, doi:10.3390/medicina60060994_

Round 1

Reviewer 1 Report

Comments and Suggestions for Authors

This article describes feasibility and safety of uniportal thoracoscopic segmentectomy.

Title: appropriate

Abstract: A short and structured abstract is presented. No change in the abstract is required

Key words: Appropriate.

Introduction: a brief introduction is given where the the conventional and modified segmentectomy have been introduced. a few previous studies have been cited. A brief note on aims and objectives can be added at the end of this section.

Methodology: total 119 patients have been used in this study. The number is adequate. However, the numbers in two groups (unidirectional and conventional) do not match. The authors must justify why they could not compare the two groups of same number of patients. inclusion/exclusion criterion can be mentioned. 

Statistics: the methods used have been briefed

Results: the outcomes of the methods used in the study have been described.

Discussion: the authors have discussed the results in this section. Comparison with the previous studies and dealing with the variations of the hilar structures and fissures and lobes has not been discussed. The authors could talk about the issues given in the following articles

George BM, Nayak SB, Marpalli S. Morphological variations of the lungs: a study conducted on Indian cadavers. Anat Cell Biol. 2014 Dec;47(4):253-8. doi: 10.5115/acb.2014.47.4.253. Epub 2014 Dec 23. PMID: 25548723; PMCID: PMC4276899.

Zhang Y, Xin W, Xu C, Yuan M, Yang G, Cheng K, Luo D. Thoracoscopic lobectomy through the pulmonary hilum approach for the treatment of congenital lung malformation. Surg Endosc. 2022 Jan;36(1):711-717. doi: 10.1007/s00464-021-08339-8. Epub 2021 Feb 16. PMID: 33591448.

Meenakshi S, Manjunath KY, Balasubramanyam V. Morphological variations of the lung fissures and lobes. Indian J Chest Dis Allied Sci. 2004 Jul-Sep;46(3):179-82. PMID: 15553206.

Limitations: The authors have agreed about the limitations in the study

Conclusions: good.

Author Response

I really appreciate your thoughtful comments for our manuscript.

The replies for your comments were below:

Comment 1) Introduction: a brief introduction is given where the conventional and modified segmentectomy have been introduced. a few previous studies have been cited. A brief note on aims and objectives can be added at the end of this section.

Ans. 1) Thank you for your suggestion. I added the sentence at the end of the introduction section.

Change in the text 1) Please see lines 48-52.

Comment 2) Methodology: total 119 patients have been used in this study. The number is adequate. However, the numbers in two groups (unidirectional and conventional) do not match. The authors must justify why they could not compare the two groups of same number of patients. inclusion/exclusion criterion can be mentioned. 

Ans. 2) Your comment is very reasonable. However, all of the performed segmentectomies in our department were enrolled in this retrospective study. Therefore, the number of patients in the two groups was different.

Change in the text 2) It would be the current form.

Comment 3) Discussion: the authors have discussed the results in this section. Comparison with the previous studies and dealing with the variations of the hilar structures and fissures and lobes has not been discussed. The authors could talk about the issues given in the following articles

George BM, Nayak SB, Marpalli S. Morphological variations of the lungs: a study conducted on Indian cadavers. Anat Cell Biol. 2014 Dec;47(4):253-8. doi: 10.5115/acb.2014.47.4.253. Epub 2014 Dec 23. PMID: 25548723; PMCID: PMC4276899.

Zhang Y, Xin W, Xu C, Yuan M, Yang G, Cheng K, Luo D. Thoracoscopic lobectomy through the pulmonary hilum approach for the treatment of congenital lung malformation. Surg Endosc. 2022 Jan;36(1):711-717. doi: 10.1007/s00464-021-08339-8. Epub 2021 Feb 16. PMID: 33591448.

Meenakshi S, Manjunath KY, Balasubramanyam V. Morphological variations of the lung fissures and lobes. Indian J Chest Dis Allied Sci. 2004 Jul-Sep;46(3):179-82. PMID: 15553206.

Ans. 3) I really appreciate your kindness. However, it might be difficult to discuss the variations of hilar structures because the number of enrolled patients was small in this retrospective study.

Change in the text 3) It would be the current form.

Reviewer 2 Report

Comments and Suggestions for Authors

This study discussed the feasibility and safety of uniportal thoracoscopic segmentectomy using a unidirectional dissection approach. This is a good topic, after all, this assumption has been controversial before. The researchers then excluded the most common sites for anatomical lung segmentectomy on left S1+2, S3, S6 and right S1, S2, S6 in the design of this study. So this has a great impact on the conclusion of this study. In addition, the researchers did not provide pictures or videos of the surgery, which I think is a pity. In our experience, unidirectional dissection approach is suitable for lobectomy, and uniportal thoracoscopic segmentectomy using a unidirectional dissection approach is suitable for very few lung segments, especially at this stage when lung segmentectomy has been used on a large scale in the clinic for less than 15 years.

Author Response

I really appreciate your thoughtful comments for our manuscript.

The replies for your comments were below:

Comment 1) The researchers then excluded the most common sites for anatomical lung segmentectomy on left S1+2, S3, S6 and right S1, S2, S6 in the design of this study. So this has a great impact on the conclusion of this study.

Ans.) I totally agree with your opinion. Therefore, it was discussed in lines 239-251.

Change in the text) It would be the current form.

Comment 2) In addition, the researchers did not provide pictures or videos of the surgery, which I think is a pity.

Ans.) Thank you for your comment.

Change in the text) It would be the current form.

Reviewer 3 Report

Comments and Suggestions for Authors

Dear authors,

Thank you for submitting your manuscript titled “Feasibility and safety of uniportal thoracoscopic segmentectomy using a unidirectional dissection approach without dissecting a fissure” for review.

This original article analyses a new surgical procedure comparing it with a classic procedure. The manuscript is well written, easy to read due to the logical thread of expression. I particularly appreciate the detailed presentation of the peculiarities of the unidirectional dissection operative technique as well as the anatomical details. This fact leads to an easy reproducibility of the mentioned technique.

However, the presentation of the results is brief. For a better understanding of the study, I suggest a detailed presentation of the results. Moreover, in the discussion chapter (which starts with a conclusion), I think that all the results should be commented step by step and compared with other results in the literature.

The conclusions are consistent with the topic.

The bibliography contains numerous recent publications. It can be enriched taking into account that 20% are self-citations.

In addition, I have identified few areas where the manuscript could benefit from further enhancements. Below are my detailed suggestions:

-          Lines 146-147 – “From February 2019 to June 2021, the tube was left in at least until postopera-146 tive day 1.” – I suggest rewording the paragraph

-          Lines 158 and 173 - the title of the subchapters is identical

-          Subchapter 3.3 - it is necessary to detail the results from Table 3

-          Subchapter 3.4 - it is necessary to detail the results from supplementary tables

-          Line 190 - for easy reading, mark the reference after the cited author. (Liu et al. [14])

I hope these suggestions will be helpful in strengthening your manuscript and better conveying the research you have undertaken. Overall, my peer review is a major revision.

Looking forward to seeing the revised version of your work.

Best regards.

Author Response

I really appreciate your thoughtful comments for our manuscript.

The replies for your comments were below:

Comment 1) However, the presentation of the results is brief. For a better understanding of the study, I suggest a detailed presentation of the results. Moreover, in the discussion chapter (which starts with a conclusion), I think that all the results should be commented step by step and compared with other results in the literature.

Ans.1) Thank you for your comment. The results section was modified, which includes details.

I think the any results were discussed in the discussion section.

Change in the text 1) Please see lines 159-181.

Comment 2) The bibliography contains numerous recent publications. It can be enriched taking into account that 20% are self-citations.

Ans.2) Your comment is reasonable. However, our group has focused on the efficacy of unidirectional dissection approach and other techniques in thoracoscopic pulmonary segmentectomy. Please accept the rate of self-citations.

Change in the text 2) It would be the current form.  

Comment 3)  Lines 146-147 – “From February 2019 to June 2021, the tube was left in at least until postopera-146 tive day 1.” – I suggest rewording the paragraph

Ans.3) Thank you for your suggestion. I revise it.

Change in the text 3) Please see line 149.

Comment 4) Lines 158 and 173 - the title of the subchapters is identical.

Ans.4) Thank you for your suggestion. I revise it.

Change in the text 4) Please see line 175.

 Comment 5) Subchapter 3.3 - it is necessary to detail the results from Table 3

Ans.5) Thank you for your suggestion. I revise it.

Change in the text 5) Please see lines 176-180.

Comment 6) Subchapter 3.4 - it is necessary to detail the results from supplementary tables

Ans.6) Thank you for your suggestion. I revise it.

Change in the text 6) Please see lines 186-192.

Comment 7) Line 190 - for easy reading, mark the reference after the cited author. (Liu et al. [14]).

Ans.7) Thank you for your suggestion. I revise it.

Change in the text 7) Please see lines 201.

Round 2

Reviewer 3 Report

Comments and Suggestions for Authors

Dear authors,

Thank you for re-submitting your manuscript titled “Feasibility and safety of uniportal thoracoscopic segmentectomy using a unidirectional dissection approach without dissecting a fissure” for review.

After the changes made, I believe that the manuscript has improved from a qualitative point of view. However, I do not agree with the wording like "Table 1 compares" and, from my point of view, the paragraphs on lines 161, 166 and 176 should be reformulated. The table is a data presentation tool and not a comparison tool.

Overall, my peer review is a minor revision.

Best regards.

Author Response

Dear Reviewer 3,

I really appreciate your thoughtful comments for our manuscript.

The replies for your comments were below:

Comment 1) After the changes made, I believe that the manuscript has improved from a qualitative point of view. However, I do not agree with the wording like "Table 1 compares" and, from my point of view, the paragraphs on lines 161, 166 and 176 should be reformulated. The table is a data presentation tool and not a comparison tool.

Ans.1) Thank you for your suggestion. They were revised.

Change in the text 1) Please see lines 161, 166 to 167, and 176.
